# Lipopolysaccharide Activating NF-kB Signaling by Regulates HTRA1 Expression in Human Retinal Pigment Epithelial Cells

**DOI:** 10.3390/molecules28052236

**Published:** 2023-02-28

**Authors:** Shengliu Pan, Min Liu, Huijuan Xu, Junlan Chuan, Zhenglin Yang

**Affiliations:** 1Natural Products Research Center, Chengdu Institute of Biology, Chinese Academy of Sciences, No. 9, Section 4, Renmin South Road, Chengdu 610041, China; 2University of Chinese Academy of Sciences, 380 Huaibeizhuang, Huairou District, Beijing 101408, China; 3Sichuan Provincial Key Laboratory for Human Disease Gene Study, Center for Medical Genetics, Department of Laboratory Medicine, Sichuan Academy of Medical Sciences & Sichuan Provincial People’s Hospital, University of Electronic Science and Technology, No. 4, Section 2, Jianshe North Road, Chengdu 610054, China; 4Department of Pharmacy, Personalized Drug Therapy Key Laboratory of Sichuan Province, Sichuan Provincial People’s Hospital, University of Electronic Science and Technology of China, No. 32, West Section 2, First Ring Road, Qingyang District, Chengdu 610072, China; 5Research Unit for Blindness Prevention of the Chinese Academy of Medical Sciences (2019RU026), Sichuan Academy of Medical Sciences and Sichuan Provincial People’s Hospital, No. 32, West Section 2, First Ring Road, Qingyang District, Chengdu 610072, China

**Keywords:** HTRA1, inflammation, LPS, celastrol, RPE

## Abstract

Inflammation and elevated expression of high temperature requirement A serine peptidase 1 (HTRA1) are known high risk factors for age-related macular degeneration (AMD). However, the specific mechanism that HTRA1 causes AMD and the relationship between HTRA1 and inflammation remains unclear. We found that lipopolysaccharide (LPS) induced inflammation enhanced the expression of HTRA1, NF-κB, and p-p65 in ARPE-19 cells. Overexpression of HTRA1 up-regulated NF-κB expression, and on the other hand knockdown of HTRA1 down-regulated the expression of NF-κB. Moreover, NF-κB siRNA has no significant effect on the expression of HTRA1, suggesting HTRA1 works upstream of NF-κB. These results demonstrated that HTRA1 plays a pivotal role in inflammation, explaining possible mechanism of overexpressed HTRA1-induced AMD. Celastrol, a very common anti-inflammatory and antioxidant drug, was found to suppress inflammation by inhibiting phosphorylation of p65 protein efficaciously in RPE cells, which may be applied to the therapy of age-related macular degeneration.

## 1. Introduction

Age-related macular degeneration (AMD) has become a major contributor to blindness in the elderly in developed countries and the third leading cause of blindness worldwide, accounting for 8.7% of all cases of permanent vision loss [1]. AMD is classified as early AMD and advanced AMD. Advanced AMD can currently be divided into geographic atrophy (dry AMD) and choroidal neovascularization (wet AMD) based on the presence or absence of neovascularization [2].

The etiology of AMD is complex. Through genome-wide association analysis it is found that complement factor H (CFH) and HTRA1 gene serves as two major genes related to the occurrence and development of AMD. More than half of AMD patients have single-nucleotide polymorphisms of these two genes [3,4]. At the same time, a considerable number of studies have proposed that overexpression of HTRA1 protein in retinal pigment epithelial (RPE) cells can induce the occurrence of AMD [5,6,7].

The pathological changes of AMD mainly occurred in choroid, bruch membrane, and retinal pigment epithelium [8]. However, the specific pathogenesis of AMD is still unclear. Studies have concluded that aging, environment, heredity, oxidative stress, metabolic disorders, and so on are risk factors of AMD. Among them, chronic inflammation is recognized as one of the most important factors. Retinal inflammation is able to induce the formation of vitreous warts, RPE and photoreceptor degeneration, and choroidal neovascularization and eventually lead to the occurrence and development of AMD. 

LPS is widely recognized to induce various inflammation including systemic inflammation, cardiac inflammation, and osteoarthritis [9,10]. Several studies have found that LPS can aggravate arthritis symptoms by stimulating the secretion of HTRA1 by macrophages in the joints and bones of mice [11]. Since both inflammation and HTRA1 expression plays important roles in the development of AMD [2,12], we intend to explore possible role of HTRA1 in inflammation with an aim to reveal potential therapeutic targets of AMD. Our results demonstrated that HTRA1 is involved in LPS-induced inflammation by regulating the expression of NF-κB. Celastrol could effectively inhibit phosphorylation of p65 to prevent activation of downstream NF- κB in RPE cells, which has potential to be applied to the therapy of AMD.

## 2. Results

### 2.1. LPS Induces HTRA1 Expression in ARPE-19 Cells

Inflammation is widely believed to involved in the processes of AMD pathological mechanisms. Elevated HTRA1 protein expression has also been detected in eyes of AMD patients [2]. Thus, we investigated the effect of LPS on the expression of HTRA1 in ARPE-19 cells. ARPE-19 cells were incubated with different concentration of LPS (0, 0.1 μg/mL, 0.5 μg/mL, 1 μg/mL), and the HTRA1 protein secreted into the medium was detected by western blot. Consistent with the results of other studies [4,11], the level of HTRA1 protein increased with the addition of LPS. The induction effect on HTRA1 expression of LPS increased with the concentration of LPS and saturated at 1 µg/mL. The amount of HTRA1 also accumulated over time with addition of LPS (Figure 1A). The mRNA of HTRA1 was also boosted after treatment of LPS. As the results of qPCR analysis showed, expression of HTRA1 rose 0.5-fold when stimulated by LPS (Figure 1B).

### 2.2. Effects of LPS on Activation of NF-κB Pathway

NF-kB signaling pathways are involved in LPS-induced inflammatory reaction in various cells [13,14]. Therefore, we tested the effect of LPS on NF-κB pathway in ARPE-19 cells. As shown in Figure 2, LPS up-regulated the expression of NF-κB at a concentration of 1 μg/mL. The mRNA level of NF-κB in LPS treated cells was 1.2-fold compared with that in control counterparts. This effect was paralleled by elevated p-p65 level after LPS induction, demonstrated by western blot analysis in Figure 2.

### 2.3. HTRA1 Regulates Expression of NF-κB in ARPE-19 Cells

Our results demonstrated that LPS prompted the expression of HTRA1 and induced activation of NF-κB pathway. In order to explore the correlation of HTRA1 and NF-κB pathway, lentivirus-mediated siRNA silence and gene expression was performed in ARPE-19 cells. Then, we verified the efficiency of HTRA1 overexpression virus, HTRA1 knockdown virus and NF-κB knockdown virus respectively (Figure 3A,C). We found an interesting phenomenon that the mRNA level of NF-κB increased in HTRA1 overexpressed cells and decreased in HTRA1 knockdown cells (Figure 3B). On the contrary, the mRNA level of HTRA1 was not affected in NF-κB knockdown cells (Figure 3D). We speculated that LPS prompted up-regulating of HTRA1, which further induced NF-κB pathway activation and inflammation.

### 2.4. Elevated HTRA1 Level Inhibits Proliferation of ARPE-19 Cells

The effect of HTRA1 on the growth is controversial. Several studies reported HTRA1 increases proliferation of ARPE-19 cells and regulated cell migration [5,8], but some research found that HTRA1 overexpression resulted in significant apoptotic changes and inhibited cell growth [4]. To determine the effect of HTRA1 on the proliferation of ARPE-19 cells, Edu was used to detect cell proliferative capacity. As shown in Figure 4, the proliferation of cells with HTRA1 overexpression is decreased, but the proliferation of cells with HTRA1 knockdown is enhanced (Figure 4). 

### 2.5. Celastrol Can Inhibit Inflammation by Inhibiting p-p65

From above results, we can know that LPS stimulation can induce the increase of HTRA1 and inflammation. Since celastrol is a kind of natural product widely used as a non-specific NF-κB inhibitor [15,16,17], we used it to observe its effect on suppressing inflammation. Different concentrations of celastrol were added to the ARPE-19 cell after incubation with 1 μg/mL of LPS. It is showed that celastrol obstructed the activation of NF-κB pathway by inhibiting the level of p-p65 obviously (Figure 5). The results showed that celastrol suppressed the phosphorylation of p65 to prevent activation of downstream NF-κB and inhibit inflammation, so celastrol may be a potential target for the treatment of AMD.

## 3. Discussion

Great progress has been made in the treatment of AMD, and therapeutic effects have been significantly improved. However, there are also some problems, such as insufficient response to treatment, drug resistance, local or systemic side effects, and high treatment cost. Therefore, it is still an urgent task to find out specific pathogenesis of AMD in order to develop better treatments. Our study focuses on whether the inflammatory reaction in the RPE cell layer will lead to the secretion of HTRA1 protein, which is coded by an AMD highest risk-related gene HTRA1.

Inflammation is a well-known risk factor of AMD due to its physiological and potentially pathological roles in RPE degeneration. Oxidative stress, accumulated metabolic waste, and other factors can induce RPE cells to secrete a large number of inflammatory factors, such as complement protein, NLRP3 inflammatory bodies, TNF-α and IL-1 β, IL-6, IL-12, IL-18, etc., thus initiating the downstream inflammatory cascade [18]. Then, inflammatory cells, including but not limited to neutrophils and macrophages, will form an inflammatory regulatory network with inflammatory factors, which accelerates the occurrence and development of AMD. The role of NF-κB, a critical component of inflammation-related signaling pathway, has gained increasing attention in the pathogenesis of AMD.

HTRA1 protein is involved in the occurrence and development of many diseases, including AMD, cerebrovascular disease, arthritis, and so on. However, the role of HTRA1 and its specific mechanism have yet to be determined. In this study, we found that the expression of the LPS-induced NF-κB pathway in ARPE-19 cells is dependent on HTRA1. The expression of HTRA1 protein in APPE-19 cell line increased after LPS stimulation. We constructed HTRA1 knockdown and overexpression viruses and transfected them into cells. We found that the expression of NF-κB was increased with HTRA1 overexpression, and NF-κB was down-regulated when HTRA1 was knocked down. After NF-κB deletion, HTRA1 expression level did not change. In addition, celastrol, an inhibitor of NF-κB, was used to treat ARPE-19 cells. Celastrol was found to block the phosphorylation of p65, thereby inhibiting LPS-induced inflammation (Figure 6). In addition, an interesting phenomenon we observed was that cell proliferation slowed down after overexpression of HTRA1 and vice versa. We speculate that in the presence of inflammation, the increased expression of HTRA1 leads to slow proliferation or even death of RPE cells, which leads to the occurrence and development of AMD [19,20,21,22]. This is consistent with the consensus that inflammation accelerates the degeneration of RPE, which further leads to the development of AMD [23].

However, all of our results were obtained from experiments conducted in REP cells. Additional data from in vivo and/or animal model experiments is required to clarify the detailed intersection of HTRA1 and inflammation in the pathogenesis AMD. HTRA1 is not widely recognized as a key component of the canonical NF-κB pathway. Whether and how HTRA1 interacts with other NF-κB pathway molecules remains unclear. It is interesting and valuable to investigate the connection of HTRA1 to other important members in NF-κB pathway.

In conclusion, we found that HTRA1 plays an important role in LPS-induced NF-κB activation. The level of downstream NF-κB can be regulated by inhibiting HTRA1. However, the specific mechanism of HTRA1 regulating the expression of the NF-κB pathway needs to be further studied.

## 4. Materials and Methods

### 4.1. Cell and Culture

ARPE-19 cells were obtained from the American Type Culture Collection (ATCC) and were cultured in standard DMEM/F-12 (Gibco, Waltham, MA, USA) containing 10% fetal bovine serum (FBS; Gibco) and 100 U/mL penicillin/streptomycin (Gibco) in a humidified incubator at 37 °C with 5% CO_2_.

### 4.2. LPS Stimulation and Celastrol Treatment

ARPE-19 cells in DMEM/F-12 grown to 70–80% of 6-cm cell dishes were cultured overnight and then treated with different concentrations (0, 0.1 μg/mL, 0.5 μg/mL, 1 μg/mL) of LPS (Solarbio, Beijing, China) for 24 h. Different concentrations (0, 0.1 μMol, 0.2 μMol, 0.5 μMol, 1 μMol, 2 μMol) of celastrol (Selleck, Houston, TX, USA) were incubated with 1 μg/mL LPS for 24 h. The expression levels of HTRA1, p65, p-p65 proteins, and mRNA in the treated cells were studied by western blot and quantitative polymerase chain reaction (qPCR). 

### 4.3. Overexpression of HTRA1 or Knockdown of HTRA1 and NF-κB

ARPE-19 cells were transduced with lentiviral vectors carrying HTRA1 gene or shRNA of HTRA1 or NF-κB at 50% confluence at a multiple of infection (MOI) of 5.

After 8 h of transduction, the cells were further cultured in DMEM/F-12 containing 10% FBS, and then screened using puromycin. When more than 90% of the transduced cells were found to strongly express GFP under the fluorescence microscope, it represented that we had established a stable shRNA inhibitor line. After 72 h of lentiviral transduction, cells were harvested and the efficiency of HTRA1 or NF-κB silencing was determined by quantitative real-time PCR.

### 4.4. Protein Extraction and Western Blot Analysis

Cells were washed with cold PBS and lysed with cold lysis buffer containing 150 mM NaCl, 50 mM Tris-HCL, 1% Triton X-100 supplemented with protease inhibitors. Cells were sonicated three times for 10 s and centrifuged at 13,000× rpm for 15 min for supernatant. Then, 20 μg of proteins was added with loading buffer, boiled at 95 °C for 5 min and loaded onto 12% polyacrylamide gels for SDS-PAGE analysis. Proteins were transferred to nitrocellulose membranes (Millipore, Billerica, MA, USA) in transferring buffer. Membranes were blocked with 8% milk for 1 h at room temperature before incubating with different primary antibodies diluted (1:2000) in 8% milk overnight at 4 °C. After 3 washes with TBST buffer for 5 min each time, membranes were incubated with secondary antibodies for 1 h at room temperature. Secondary antibodies were diluted (1:10,000) in 5% TBST. After 3 washes with TBST for 5 min each, the proteins on the membranes were visualized using enhanced chemiluminescence reagents and autoradiography (Bio-Rad). Protein bands were quantified by densitometry using ImageJ software. Levels of the proteins of interest were normalized to GAPDH.

### 4.5. Quantitative RT-PCR Analysis

Total RNA was extracted from ARPE19 with RNeasy Mini kits (QIAGEN, Hilden, Germany), and 1 μg total RNA was reverse transcribed with EasyScript Two-Step RT-PCR SuperMix 15 (TransGen Biotech, Beijing, China) following the manufacturer’s instructions. cDNA was amplified using TransStart Tip Green qPCR SuperMix (TransGen Biotech, Beijing, China) in a 7500 Fast Real-17 Time PCR System (Thermofisher, Waltham, MA, USA). Primers were listed in Table 1.

### 4.6. EdU Labeling of ARPE-19 and Immunofluorescence Staining

ARPE19 cells were seeded on 5 µg/mL human fibronectin protein-coated slides in 24-well plates (Corning, NY, USA). Then, ARPE19 cells were treated with 10 µmol/mL EDU for 4 h, and EDU-positive cells were visualized by subsequent staining with an EDU Cell Proliferation Kit with Alexa Flour 594 (BeyotimeClickTM, JiangSu, China) according to the manufacturer’s instructions. After being fixed in 4% PFA in PBS at room temperature for 20 min, the slides were rinsed twice with PBS and permeabilized with 0.2% Triton X-100 containing fetal bovine serum 5% (FBS) for 20 min. Nuclei were then stained with DAPI for 1 h before harvest (1:1000 dilution; Cell Signaling Technology). Images were captured under an optical microscope (Zeiss, Germany).

### 4.7. Statistical Analysis

Every experiment was repeated at least three times for consistency. The experimental data were expressed as means (x) ± standard deviation (SD) and analysed with software GraphPad Prism 8 for group-wise comparisons and statistical analyses. T-test was used for comparison between two groups. Differences were considered statistically significant when *p* < 0.05. GraphPad Prism software was used for the calculations.

## Figures and Tables

**Figure 1 molecules-28-02236-f001:**
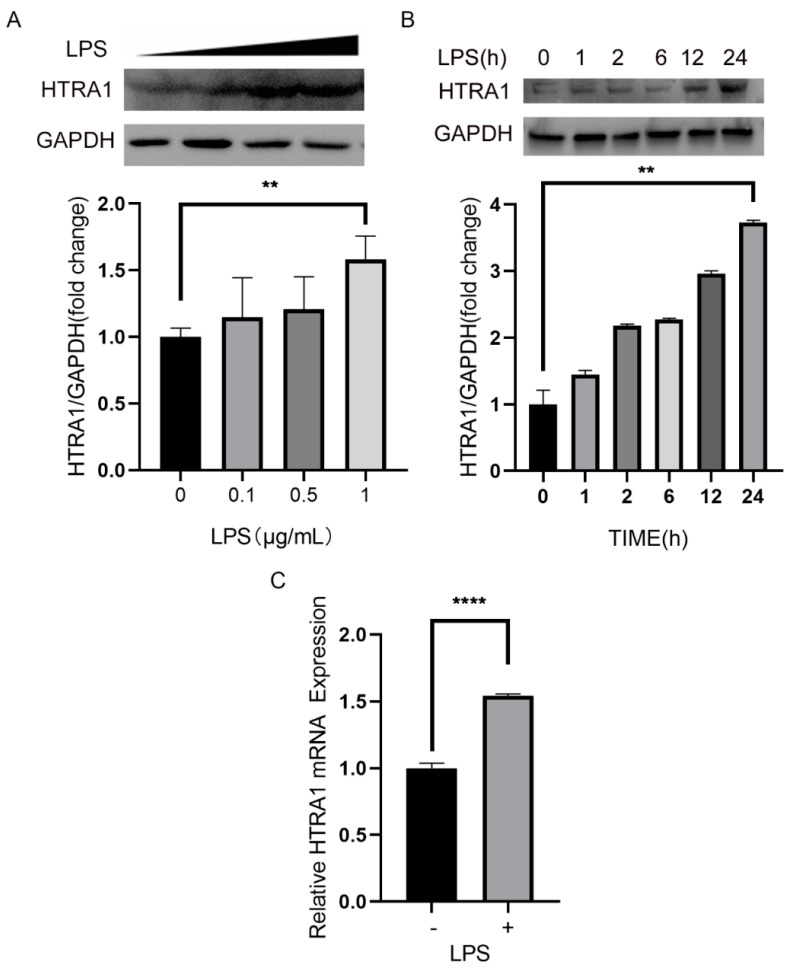
HTRA1 expression in ARPE-19 cell line after LPS stimulation. (**A**) Western blot analysis of HTRA1 expression in ARPE-19 cell line stimulated by different concentrations of LPS for 24 h. ** *p* < 0.01. (**B**) Western blot analysis of HTRA1 expression in ARPE-19 cell line stimulated by 1 µg/mL LPS for different time. ** *p* < 0.01 (**C**) Changes of HTRA1 mRNA after stimulation of 1 µg/mL LPS for 24 h. **** *p* < 0.0001.

**Figure 2 molecules-28-02236-f002:**
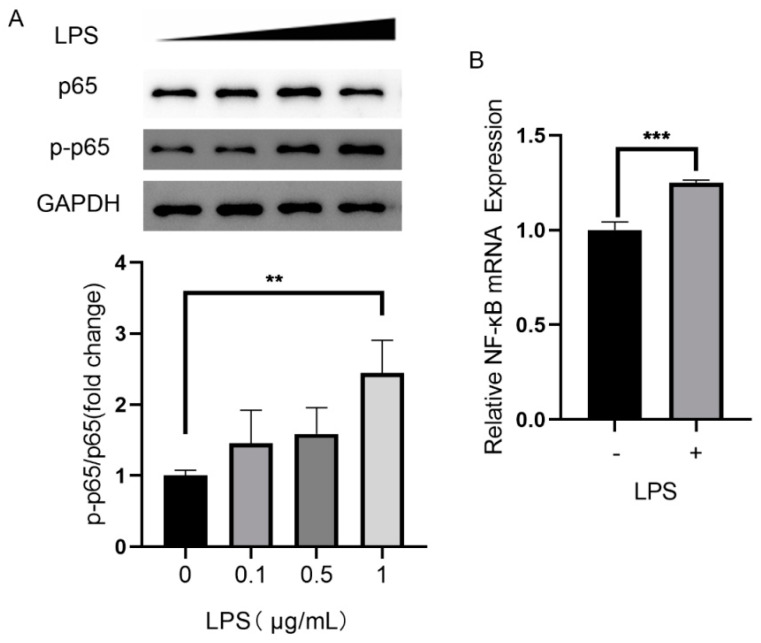
Effect of LPS on the expression of NF-κB and p-p65 protein in ARPE-19 cells. (**A**) Protein level of p65 and p-p65 with LPS treatment. ** *p* < 0.01. (**B**) mRNA level of NF-κB after LPS stimulation. *** *p* < 0.001.

**Figure 3 molecules-28-02236-f003:**
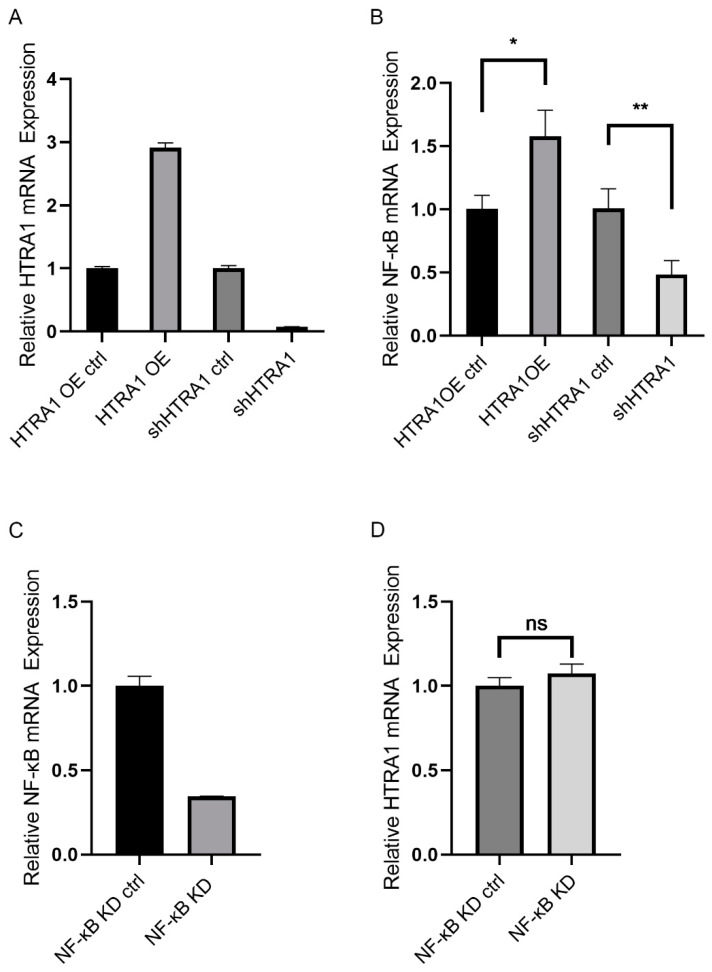
Role of HTRA1 in LPS-induced NF-κB expression in ARPE-19 cells. (**A**) Verification of efficiency of HTRA1 knockdown and overexpression; (**B**) Relative mRNA level of NF-κB in HTRA1 knockdown and overexpression cells; (**C**) Verification of efficiency of NF-κB knockdown; (**D**) Relative mRNA level of HTRA1 in NF-κB knockdown cells. * *p* < 0.05, ** *p* < 0.01, ns = no significance.

**Figure 4 molecules-28-02236-f004:**
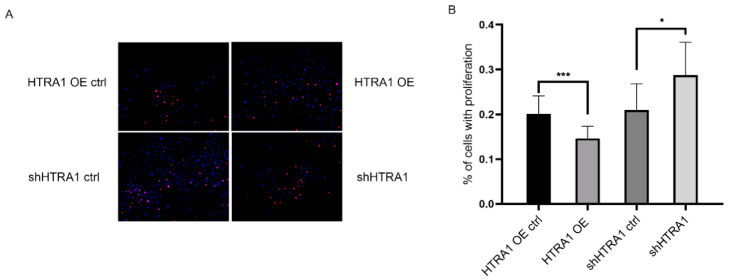
Effect of HTRA1 on the proliferation of ARPE-19 cell line. (**A**) Representative fluorescent image of EDU labeled ARPE-19 cell of HTRA1 overexpression and knockdown group. (**B**) Statistical result of percentage of proliferating cells. *** *p* < 0.001, * *p* < 0.05.

**Figure 5 molecules-28-02236-f005:**
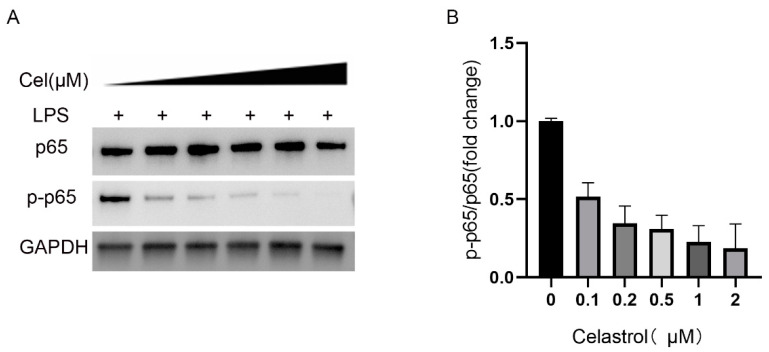
Treatment of ARPE-19 cell line with different concentrations of celastrol. (**A**) Protein level of p65 and p-p65 with celastrol treatment. (**B**) Fold change level of p-p65/p65 after celastrol stimulation.

**Figure 6 molecules-28-02236-f006:**
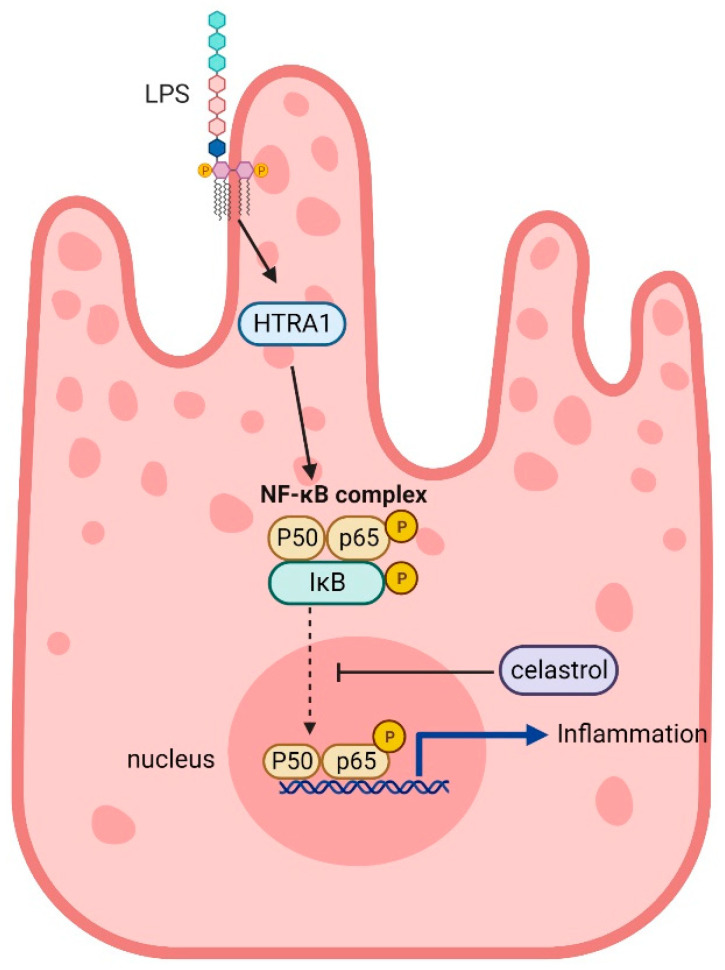
Schematic diagram demonstrating the role of HTRA1 in LPS-induced inflammation in RPE cells and potential therapeutic application of celastrol in AMD.

**Table 1 molecules-28-02236-t001:** Primers for Quantitative RT-PCR analysis.

Primer Name		Squence (5′-3′)
HTRA1	reverse forward	GTCACTCACGTCCAGCAAAG TTCGACCACCAGAGTTCCTT
NF-κB	reverse forward	AGGATTTCGTTTCCGTTATGT CTTGTTCTTCAGAATGGGAGTCC
GAPDH	reverse forward	CTGACTTCAACAGCGACACC GTTGCTTAAACCGATGTCGT

## Data Availability

All the data is included in this article.

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
