# Peer review of "Lipopolysaccharide Activated NF-kB Signaling by Regulating HTRA1 Expression in Human Retinal Pigment Epithelial Cells"

_molecules, 2023, doi:10.3390/molecules28052236_

Round 1

Reviewer 1 Report

Pan et al presents an interesting paper about the effect of HTRA1 on NF-kB signaling activation in LPS induced inflammation in Human Retinal Pigment Epithelial Cells. Assessment of HTRA1 protein expression in ARPE-19 cells after incubation with different concentration of LPS was conducted and also the effect of LPS on NFKB signaling was evaluated. The effect of HTRA1 knockdown and overexpression on NF-kB was assessed.

The experiments appear well conducted and the writing is concise. In general experimental results support conclusions and statistical analysis seems appropriate. However, some minor and major concerns have to be addressed:

Minor comments:

1- I suggest that the full term of the abbreviated HTRA1 to be addressed in the abstract text

2- I suggest that the full term of the abbreviated CFH gene to be addressed in the introduction section (line 41)

3- In the legend of figure 1 (line 76), I think it will be better to label the panel presented the effect of 1µg/mL of LPS for different time as panel (B) with more clarification for its explanation and the panel (B) changed to (C)

4- Line 119, there is a missed word “non- specific inhibitor of Nf-kB” the word inhibitor is lost

Major comments:

1- Line 123 and 124 “The results revealed therapeutic potential of celastrol to AMD because inflammation and HTRA1 elevation plays an important role in AMD pathology” this sentence better to be amended or deleted as the result here concerning effect of celastrol on NF-kB and celastrol effect on HTRA1 not assessed.

2- Line 153 in the discussion part need to be revised

3- In material and methods section (line 179), the sentence “Celastrol was removed after incubation with cell culture medium for 30 minutes” need to be clarified, celastrol was added to which group and how it was removed from the culture medium and why?

Reviewer 2 Report

This study report the HTRA1  role in LPS induced NF-κB activation. the MS is well structured, the figures are adequately illustrated. I offer some recommendations to improve this MS.

General comments:

- Introduction: authors must give more details on the problematic of this study.

- Materials and methods What is the origin of the lipopolysaccharides used in this study? Authors must mention the different concentrations used for each treatment.

- "13,000×rpm"  instead  "13,000×g"
